

# Nest-site selection, reproductive ecology and shifts within core-use areas of Black-necked Cranes at the northern limit of the Tibetan Plateau

Lixun Zhang[1,2], Bei An[3], Meilin Shu[1] and Xiaojun Yang[4]

[1] School of Life Sciences, Lanzhou University, Lanzhou, China
[2] Gansu Key Laboratory of Biomonitoring and Bioremediation for Environmental Pollution, Lanzhou, China
[3] Basic Medicine Sciences, Lanzhou University, Lanzhou, China
[4] State Key Laboratory of Genetic Resources and Evolution, Kunming Institute of Zoology, Chinese Academy of Sciences, Kunming, China

## ABSTRACT

We investigated population dynamics, breeding pairs, breeding habitat selection, nest density, distance between neighboring nests, nest survival, reproductive success, and recruitment rate for Black-necked Cranes (BNC, *Grus nigricollis*) during 2013–2015 in Yanchiwan National Nature Reserve (YCW), Gansu, China. Numbers of BNC and breeding pairs remained relatively stable at around 140 individuals and 40 pairs. Recruitment rates ranged from 15.7% to 25.8%. The average nest distance was 718.66 $\pm$ 430.50 m (2013), 1064.51 $\pm$ 323.99 m (2014) and 534.99 $\pm$ 195.45 m (2015). Average nest survival rate, hatching success, and breeding success of all 29 nests were 65.56 $\pm$ 5.09%, 57.04 $\pm$ 6.12% and 32.78% $\pm$ 2.55. Water depth, water body area, and distance to land were positively related to nest survival, while disturbance level showed a negative relationship. However, nest site selection of BNC was determined by habitat type, disturbance and water depth. BNC often foraged in mudflats and freshwater marsh but seldom foraged in saline-alkali wet meadows due to food density and quantity in April, the month when BNC choose nest sites. Conservation strategies based on habitats should consider ecological factors that may not be well predicted by nest site selection. Shifts within core-use areas from satellite tracking of BNC demonstrated that maintaining populations demands that conservation areas are large enough to permit breeding BNC changes in space use. Our results are important for conservation management and provide quantitative reproductive data for this species.

Corresponding authors
Lixun Zhang, zhanglixun@lzu.edu.cn
Xiaojun Yang, yangxj@mail.kiz.ac.cn

## INTRODUCTION

The Black-necked Crane (BNC, *Grus nigricollis*) is currently listed as a globally threatened species (IUCN: Vulnerable, *BirdLife International, 2012*) due to widespread wetland loss and agricultural development (*Harris & Mirande, 2013*). It is the only crane species that breeds completely on the high-altitude wetlands of Qinghai-Tibetan Plateau (*Li & Bishop, 1999*). Harsh and variable conditions of alpine habitats are particularly challenging for

birds to breed in and impose additional constraints in terms of nest-site selection and nest survival (*Macdonald et al., 2015*). Several studies have documented BNC nest characteristics (including nest length, nest width and nest height; *Wang et al., 1989*; *Dwyer et al., 1992*; *Wu et al., 2009*) and nest site characteristics (*Kuang et al., 2010*; *Wu et al., 2009*). Only one study has reported BNC nest site selection in Ruoergai (*Wu et al., 2009*). Understanding habitat selection and reproductive success is crucial to protect and recover threatened species (*Zhu et al., 2012*). So far, Black-necked Crane chick survival rate has only been discussed in Longbao National Nature Reserve (*Farrington & Zhang, 2013*). No estimates of nest survival or reproductive success are available for the species. *Duangchantrasiri et al. (2016)* suggested that estimations of both abundance and demographic factors that drive threatened species responses are vital. Studies carried out in Zhigatse Prefecture (*Bishop, Tsamchu & Li, 2012*), in the Altun Mountain Reserve (*Zhang et al., 2012*) and Longbao National Nature Reserve (*Farrington & Zhang, 2013*) have provided valuable information on numbers and distribution of BNC during the breeding season or wintering season. Nevertheless, little information is available on BNC in Yanchiwan National Natural Reserve (*Zhang et al., 2014*). Knowledge of how animals choose habitat and foraging resources is a vital element of basic and applied ecology (*Chudzińska et al., 2015*). However, so far only *Kuang et al. (2010)* documented that BNC foraging habitat selection in northern Tibet.

To develop effective conservation measures, it is crucial to understand the spatial requirements of Black-necked Crane pairs. However, only one attempt to estimate the distance between neighboring Black-necked Crane nests has been made (*Lu, Yao & Liao, 1980*). *Farrington & Zhang (2013)* speculated that breeding cranes might remain on or near their nesting territories until just before leaving for autumn migration. To our best knowledge, there has been no estimate size of breeding BNC nesting territories. Further, no studies have used satellite tracking to determine shifts in core-use areas for breeding BNC pairs.

Here we investigated the northern most breeding population of Black-necked Cranes at YCW. Our concern was that peripheral populations such as the cranes at YCW might be essential as species adapt to directional changes in climate or decreased habitat quality. The objectives of our study were to: (1) provide data on population dynamics of BNC and breeding pairs, nest density, distance between neighboring nests, and recruitment rates of the BNC; (2) supply reproductive success data on this species; (3) provide data on shifts in core-use areas from satellite tracking of breeding BNC families; (4) investigate quantity and density of food resources in different foraging habitats when BNC choose their nest sites in April; and (5) document basic nest and nest site characteristics and determine whether BNC select nest sites based on specific habitat components.

## METHOD

### Study area
Our breeding ecology study was carried out from March 2013 to June 2015 in Yanchiwan National Nature Reserve (YCW, 38°26′∼39°52′N, 95°21∼97°10′E). The reserve covers an area of 13,600 km$^2$ and is located on the northern edge of the Qinghai–Tibetan Plateau in

the western Qilian Mountains. The YCW is situated in a valley and consists of 4.9% marsh, 2.4% permanent pond and riverine wetland, 2.4% seasonal riverine wetland, 0.1% glacier wetland, and 90.2% grassland, which is partially fenced and used collectively by groups of herdsmen families. Plants are short and small, many of which belong to alpine cushion vegetation. Elevations range from 2,600 to 5,483 m in a broad mountain valley flanked by ridges.

YCW is characterized by a continental arid and semi-arid climate. Precipitation in YCW ranges from 33.5–40.4 mm during the April–June period and average temperatures are 5.67–6.49 °C during three monitoring breeding seasons (April–June). Snowfall can be recorded any time. Water supplies are dependent on snowmelt from Qilian Mountains, meltwater from glaciers, precipitation, surface runoff and a few freshwater springs. Water drains into the Shule, Danghe, and Yulinhe Rivers.

## Population survey

We established 3 routes (driving distance about 33.1 km, 22.6 km and 60.2 km; Fig. S1) and appropriate observation points on hills (19 set survey points), so observers could view all areas in the wetland. Using binoculars (8 × 42) and spotting scopes, 2–3 observers conducted censuses by road surveys and ground searches to locate and count cranes. We thoroughly scanned for cranes within valleys, including meadows, ponds, marsh, riverine and land areas. Observers and all methods stayed the same over the survey period. All territorial locations were determined by observations of territorial behavior including ritualized threats and pecking and chasing invaders (*Yang et al., 2007*) and by satellite tracking (for those birds with tracking devices). Territories were also identified by observing locations with active single or paired Black-necked Cranes using a telescope and ascertained by crane footprint tracks. GPS locations were taken to specify the location of all territorial pairs or single cranes. The distances between neighboring nests were measured by Google Earth Pro (Version 7.0, Google Inc. 2012) and presented as average level ± SD. Nest density estimates of Black-necked Cranes for YCW area were computed as the number of pairs per km$^2$ using distance sampling methods (*Buckland et al., 2001*). The "adult" category included both mature breeding cranes and nonbreeding cranes that were always observed as a pair. The "subadult" category included returning nonbreeding, usually young individuals, which were always present in a flock. The "chick" category included chicks hatched in 2013, 2014 and 2015, which were easily distinguishable from adults and subadults in the hatching year by their size, head and neck plumage. Recruitment was defined as frequency of chicks/100 cranes (adults, subadults, and chicks; *Bishop, Tsamchu & Li, 2012*).

Two chicks were rescued because of illness, held for 10 medical days in a work station, and then released on their nest territories. In order to monitor the recovery of the chicks and evaluate the reliability of our territory observation method, we attached satellite transmitters (platform transmitter terminals, PTTs, ModelAnti–GT0325; Blue Oceanix Inc., China; weight: 30 g; GPS–orientation; GSM transmission 1 point/2 h). Transmitters were mounted on the backs with Teflon-treated ribbons as described by *Higuchi et al. (2004)*. The harness and PTT weighed about 40g, which is approximately 2% of the body

weight of a Black-necked Crane chick (mean = 3,390 g, $n = 2$ chicks). The two birds recovered and migrated on 8 and 14 November 2015. Home range was defined as total area occupied by an individual and was analyzed in Google Earth Pro (Version 7.0, Google Inc. 2012) using its polygon tool. Shift of roosting habitat was calculated by counting the times a BNC chick rested at different sites between 00:00:00–6:00:00, by this method also identifying their roosting sites, the longest distance from their nests, and their center of activity spots. We also calculated their daily home ranges.

## Reproductive performance

We located 29 nests (an average of almost 10 nests per year) from April 2013 to June 2015 and checked nests periodically (median interval = 4 d) until the nest failed or chicks fledged. To control for resampling and possible seasonal shifts in nest site characteristics due to changing vegetation, we excluded renests. We visited nests around noon when parents left for foraging or other activities and we avoided visiting nests during the laying period. Nest location (from GPS) and nest status were recorded. Nest construction and incubation behaviors were recorded by infrared cameras (FC-5210 mm, MMS Trail Camera, Shenzhen Baird Share Co., Limited). During the nesting period, information was recorded on the number of young, nest construction, and nest materials. Fates of nests and all eggs in clutches were verified by record (Figs. S2D and S3). Nests were considered as "successful" if one chick or more chicks were confirmed to have been produced. Nest survival rate was calculated as the percentage of the number of successful nests from total number of nests initially found. Number of chicks fledging from total number of eggs laid provided an estimation of breeding success (*Mukherjee, Borad & Parasharya, 2002*). Hatching success was calculated as the probability that eggs present at hatching time actually produced young.

## Foraging resource density and quantity

The diets of BNC consist primarily of roots and tubers, insects, snails, shrimp, fish, small birds and rodents (*Bishop, 1996*). For this study we investigated tubers (including *Carex atrofusca*, *Carex orbicularis*, *Carex microglochin*), which are the only available foods during April, the month when nest sites are selected. Foraging habitats of BNC in YCW were mainly mudflats, marshes and sometimes saline-alkali wet meadows (Fig. 1). We randomly placed a number of 1 m × 1 m quadrats within the three foraging habitats to investigate food diversity and quantity from 10 to 30 April in 2014. We investigated 116 food resource samples for estimating tuber density, and collected 88 samples for estimating tuber quantity (density of tubers and fresh individual weight).

## Nest and nest site characteristics

BNC nests were built on elevated grassy islands or aquatic vegetation within wetland habitats. Nest parameters were measured (in cm). Six site characteristics were recorded to determine their influence on nest-site selection: (1) disturbance—based on videos taken by infrared cameras, disturbance was rated as (a) strong: accessible as close as 10–20 m for humans, livestock or dogs, resulting in eggs preyed or nests destroyed (Fig. 2A and Figs. S2A–S2C), or (b) weak, inaccessible due to a water barrier for livestock or dogs, and BNC remained at the nest even when intruders attempted to approach (Fig. 2B); (2) water

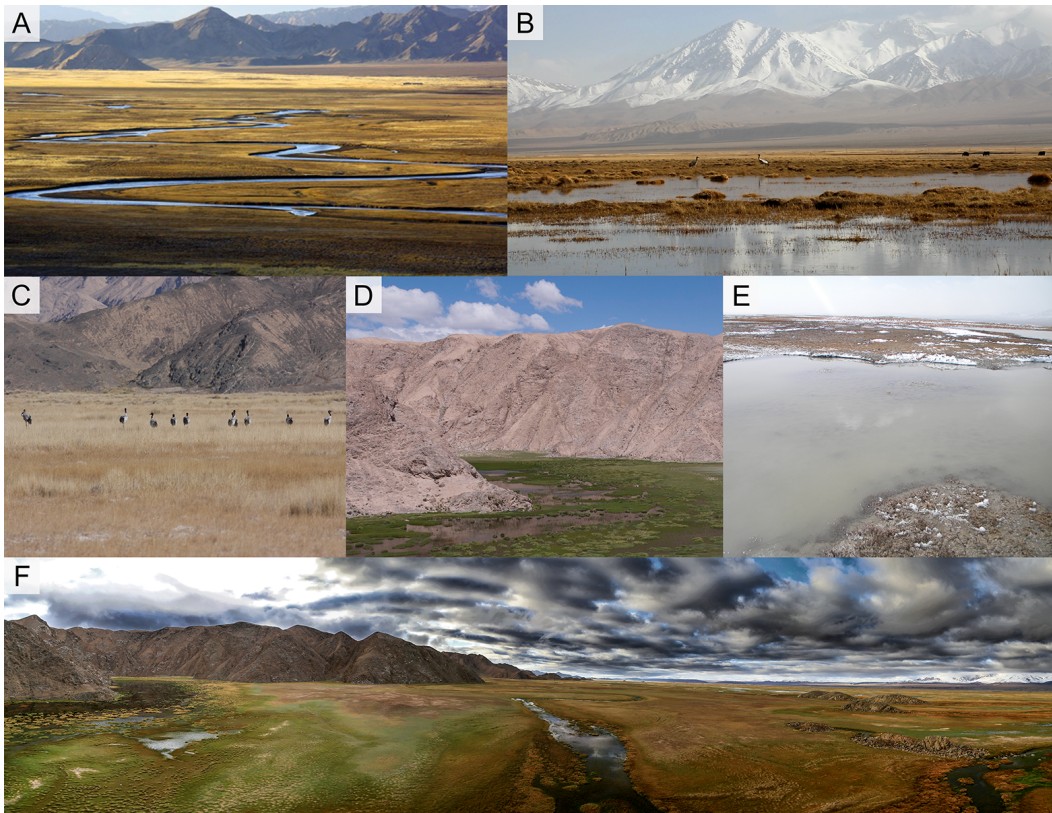

**Figure 1  Different wetland landscapes for Black-necked Crane in Yanchiwan National Nature Reserve, Gansu, China.** (A) Riverine wetland, (B) freshwater Marsh, (C) saline-alkali wet meadow, (D) pond, (E) mudflat and (F) an overview of Yanchiwan National Nature Wetland. In five habitats, (A), (B) and (D) are nest habitats. (B), (C) and (E) are foraging habitats. Figure (A) photo from Yanchiwan National Nature Wetland Authority. Figures (B)–(F) photos taken by LX Zhang.

body area (measured in ArcGIS, Version 10.2, ESRI, Redlands, CA, USA), water body area categorized as a <500 m² or b >500 m². (3) water depth (in cm, averaged from four samples taken from four cardinal directions at 1 m distance from the nest edge (*Dwyer et al., 1992*); (4) distance to the nearest land (in m) measured by infrared distance meter; (5) distance to the nearest hill and rated as a = distance from nest to the nearest hill greater than 100 m) and b = distance from nest to the nearest hill less than 100 m; and (6) nest habitat type based on hydrological and topographical characteristics (riverine wetlands are permanent, slow-flowing waters having a well-developed flood plain, while marshes and ponds are water body areas larger than 500 m², Fig. 1).

## Data analysis

Differences of spacing between nests were examined using one-way ANOVA. For nest site selection, all six site factors were subjected to Factor Analysis (FA) to determine which variables were driving the trends of nest distribution. Before the comparison with *t*-tests or nonparametric tests, a one-sample Kolmogorove-Smirnov test was used to determine whether the data were normally distributed (if so, parametric *t*-tests were used; otherwise, nonparametric tests were used). Nest characteristics among three habitats,

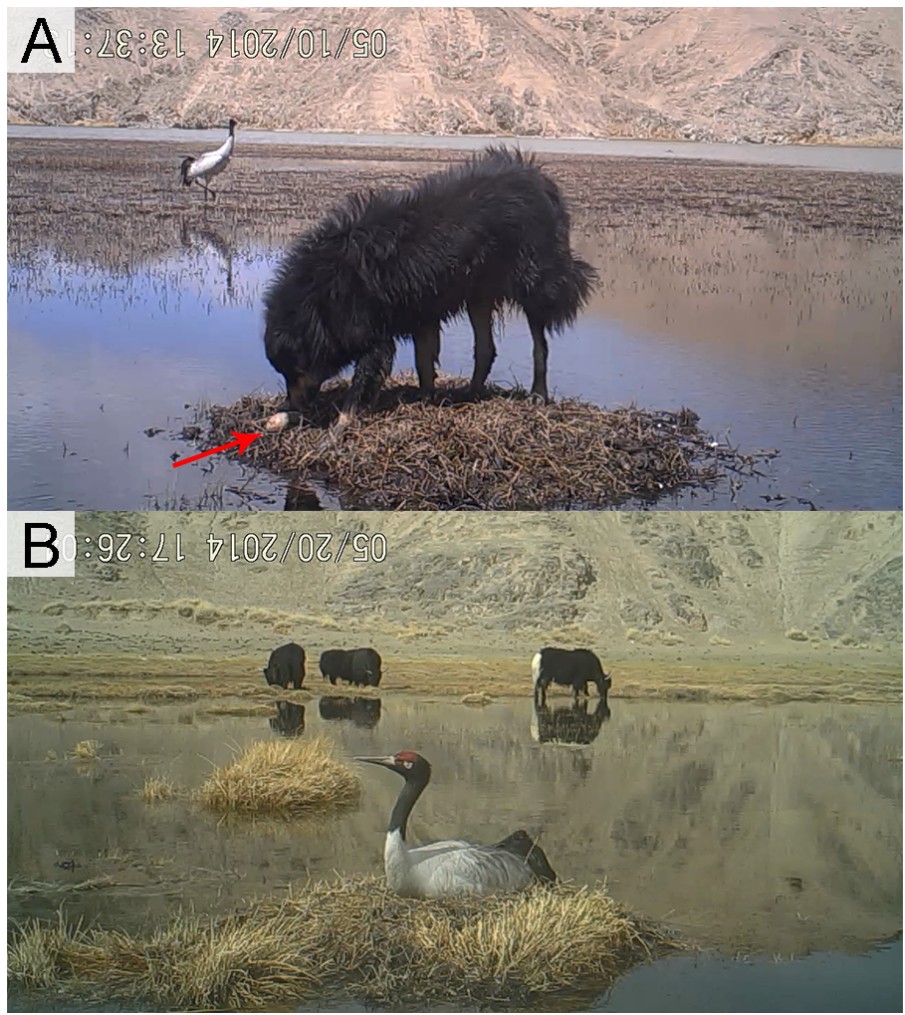

**Figure 2** **Different disturbance levels to Black-necked Crane nests.** (A) Strong disturbance: accessible for livestock or dogs, resulting in egg predation and nest destruction and (B) weak disturbance: inaccessible for livestock or dogs, and crane behavior does not change by their presence (all photos captured from video footage at nests of Black-necked Cranes).

between two nest types, and nest sites among three habitats were examined using a one-way ANOVA test, Kruskal–Wallis test or Mann–Whitney $U$ test. Nest characteristics and habitat characteristics were compared between successful nests and unsuccessful nests by one-way ANOVA test or Mann–Whitney $U$-test. Differences of nest success and nest survival between haystack nests and ground nets were analyzed using Mann–Whitney $U$ test. Kruskal–Wallis test was used in the comparison of nest survival rate among the three nesting habitats. The density and quantity of tubers in different foraging habitats were compared with the Kruskal–Wallis test. All the statistical analyses were performed by the software SPSS (version 22.0, IBM 2013). Results were given as mean ± SD, and all significance values are at 0.05 based on two-tailed tests. All statistical graphs were made in software Origin (version 9.0, Origin Lab Corporation, USA) and pictures were processed in Adobe Photoshop CS6 (version 13.0, Adobe Systems 2012).

### Ethical note

All data collected as part of this study were approved by the Lanzhou University Institutional Animal Care and Use Committee (approval numbers: SCXK-GAN-2013-0003). Field work was approved by authority of the Forestry Department of Gansu Province (approval number: 201009).

## RESULTS

### Population survey

Surveys were conducted at YCW 56 times from 30 March 2013 to 10 November 2015 (Table S1). BNC arrived in YCW from late March to mid-April. Territories were typically selected and established between 15 and 25 April. Most nests were monitored from initiation (onset of incubation), dates ranged from 20 to 30 April. Eggs were usually laid during the first two weeks of May.

BNC populations remained relatively stable from June and October during our three monitoring years (Fig. 3). In 2015 the first four chicks were observed on 30 May, reaching peak numbers of 42 chicks on 20 June. Chick recruitment in October 2015 was 15.8% (20 chicks/127 total cranes), substantially lower than the 25.7% (38 chicks/148 total cranes) recorded during October 2014. The greatest number of BNC observed was 138 on 15 July 2015. Cranes started to migrate on 10 October and the last crane departed on 10 November (Fig. 4). Satellite-tracking data indicated breeding pairs were present in their territory most of the time. The average daily home ranges of the two chicks prior to migration were 0.55 km$^2$ and 1.55 km$^2$ (Fig. S4), respectively. Roost sites for both chicks shifted throughout the season. The maximum distances from the nest site recorded for a roost site were 3.22 km and 1.29 km, respectively for these two chicks.

The number of breeding pairs ranged from 40 to 46 (40, 2013; 46, 2014; 42, 2015) in YCW. The average nest density was 1 nest/10–12 km$^2$. Spacing between nests was significantly different (one-way ANOVA, $F_{2,27} = 4.53$, $P = 0.029$) for the three years. The average between-nest distance was 718.66 ± 430.50 m (2013), while 1064.51 ± 323.99 m (2014) and 534.99 ± 195.45 m (2015).

### Reproductive performance

Across the 3 years, 23 of 29 monitored nests were successful. Hatching success in 2013, breeding success and nest survival rate in 2015 were the highest during three monitoring years (Table 1). Breeding success was low compared to nest survival rate and hatching success (Table 1). Average nest survival rate, hatching success, and breeding success of all 29 nests were 65.56 ± 5.09%, 57.04 ± 6.12% and 32.78% ± 2.55%.

### Foraging resource density and quantity

Three species of tubers *Carex atrofusca*, *Carex orbicularis* and *Carex microglochin* occurred in all three foraging habitats. The average tuber densities in the three foraging habitats were significantly different (Kruskal–Wallis, $\chi^2 = 28.41$, $P < 0.001$; Fig. 5). Tuber quantity in three foraging habitats was significantly different (Kruskal–Wallis, $\chi^2 = 13.12$, $P = 0.001$; Fig. 5).

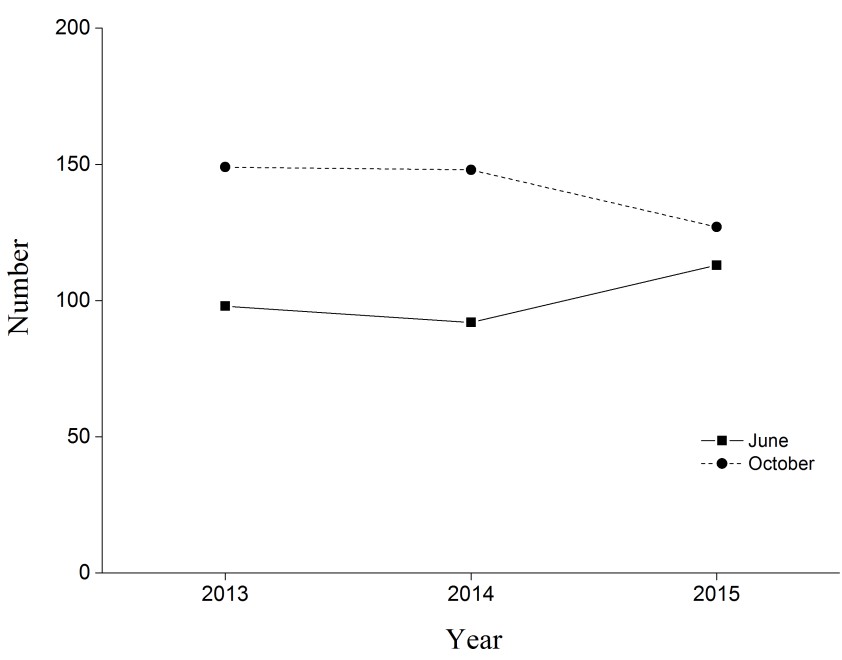

**Figure 3  Black-necked Crane census in Yanchiwan National Nature Reserve, Gansu, China, in June (solid line) and October (dash line) 2013, 2014 and 2015.**

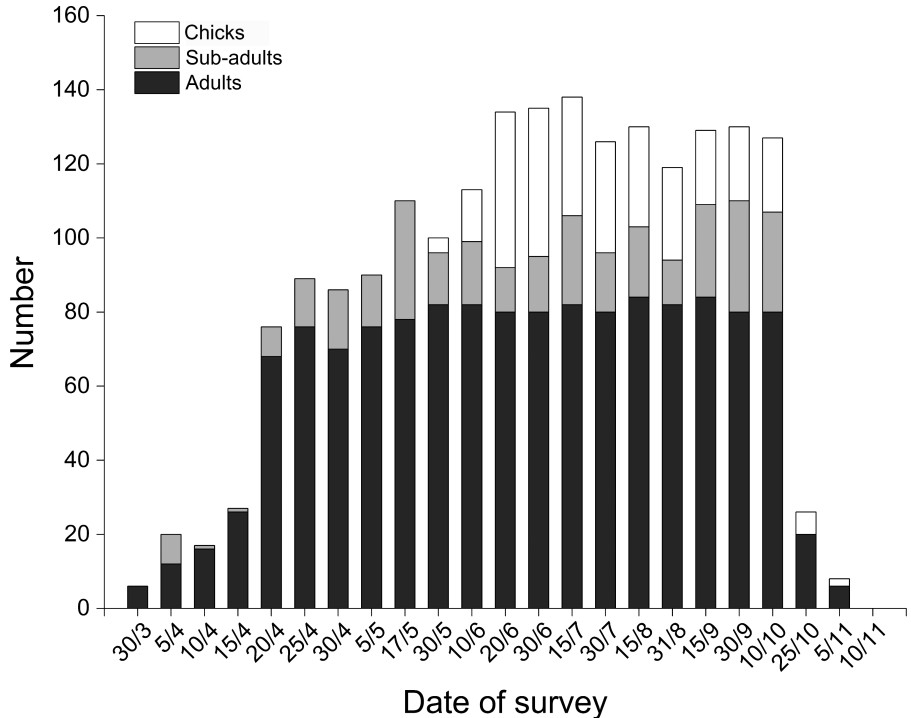

**Figure 4  Distribution of Black-necked Cranes by age class and date in Yanchiwan National Nature Reserve, Gansu, China between 30 March and 10 November 2015.** Black-necked Cranes were divided into three age classes, adults (dark), sub-adults (grey) and chicks (white).

**Table 1**  Variations in nest fates and reproductive success of the Black-necked Cranes during our monitoring years in Yanchiwan National Nature Reserve, Gansu, China.

|  | 2013 | 2014 | 2015 |
| --- | --- | --- | --- |
| Nests | 9 | 10 | 10 |
| Number of egg laid | 18 | 20 | 20 |
| Number of egg hatched | 11 | 10 | 12 |
| Number of chicks migrated | 6 | 6 | 7 |
| Nests destroyed by predators | 2 | 2 | 2 |
| Nests with eggs addled or infertile | 1 | 2 | 1 |
| Nests with at least one egg hatched | 6 | 6 | 7 |
| Hatching success | 61.11% | 50.00% | 60.00% |
| Nest survival rate | 66.67% | 60.00% | 70.00% |
| Breeding success | 33.33% | 30.00% | 35.00% |

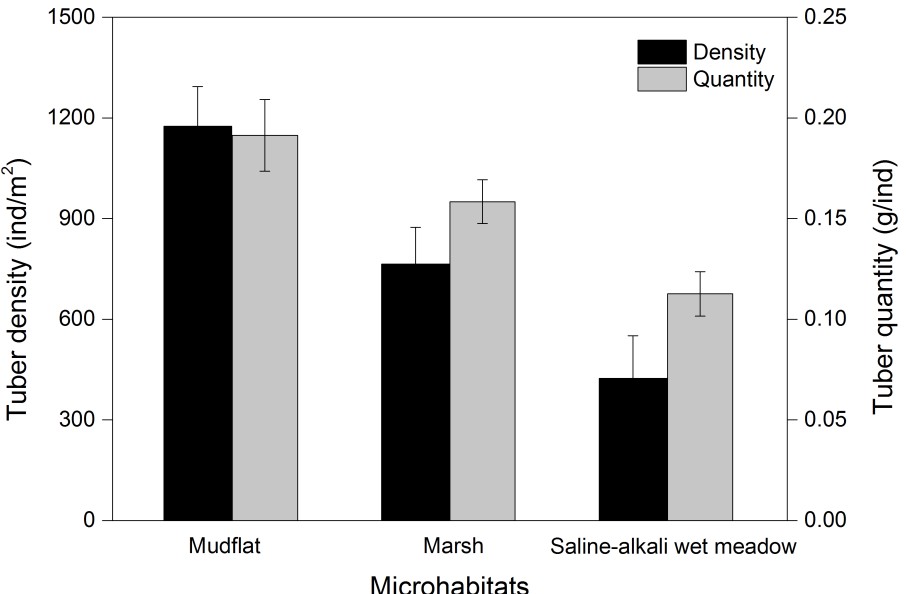

**Figure 5**  The tuber density (ind/m$^2$) and quantity (g/ind; mean ± SD) of three foraging habitat types of Black-necked Cranes in Yanchiwan National Nature Reserve, Gansu, China.

## Nest characteristics and their influences on nest survival

Nests were classified into two types (haystack nest and ground nest) depending on the nest materials and construction process (Figs. 6A and 6B). The breeding pairs that built haystack nests would select nest sites early and always attempted to raise the nest platform by adding fresh nest material above the water. They were apparently more active than those pairs using ground nests, and moved more from one place to place within the territory. Haystack nests would take around 7 to 10 days to build. Haystack nests needed to be constructed and repaired before and during incubation, which were categorized as energy consuming nests (*Wang et al., 1989*; *Wu et al., 2009*). Nest materials used in construction were often residual vegetation from the previous year (Fig. 6A). Ground nests, which were

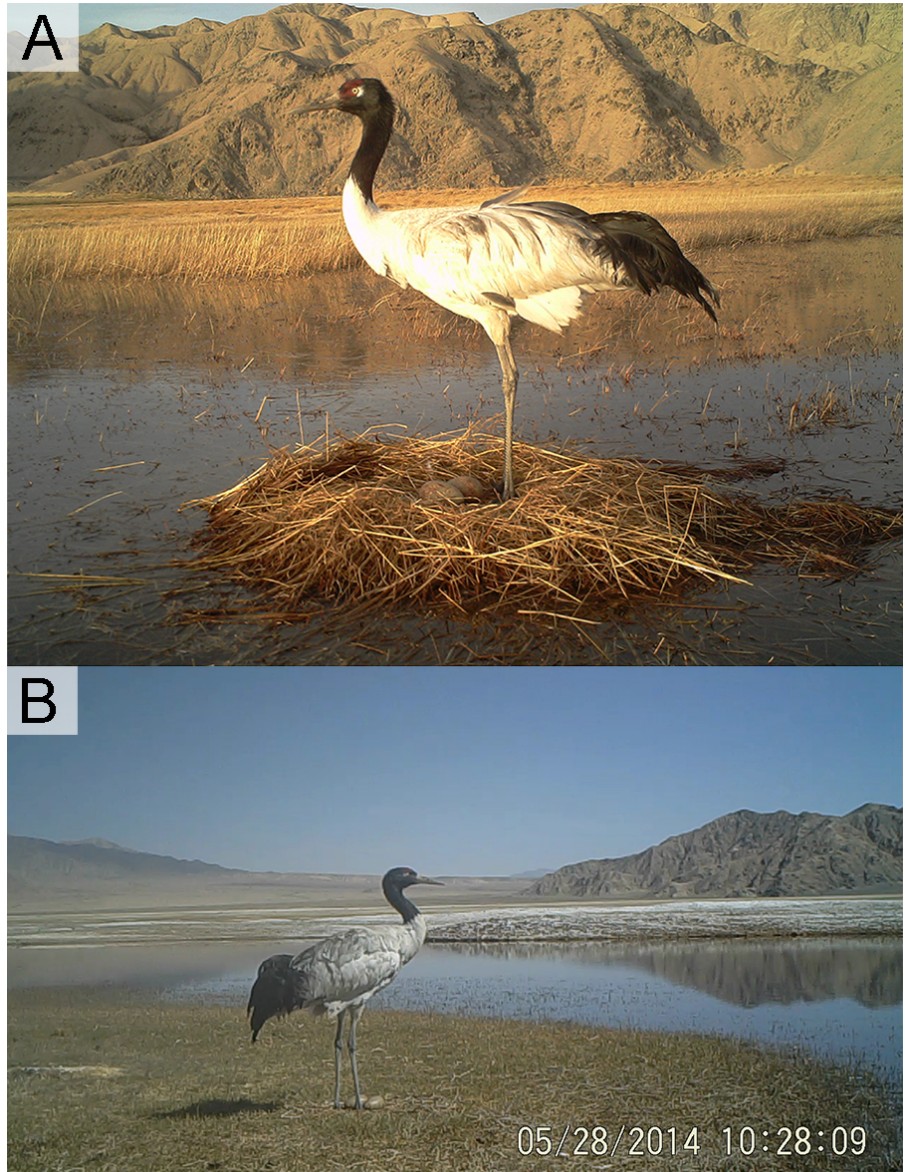

**Figure 6** **Nest types of Black-necked Cranes in Yanchiwan National Nature Reserve, Gansu, China.** (A) Haystack nest was built up with vegetation and (B) ground nest was built with minimal vegetation added (both from still photos captured from video footage at nests of Black-necked Cranes).

also called island nests, lay directly on the platform without little material added (Fig. 6B). The breeding pairs for ground nests spent little time building their nests and tended to build their nests later than pairs building haystack nests. Ground nests were categorized as non-energy consuming.

The length, width and height of haystacks were larger than those of ground nests (Table 2). However, the water depth surrounding ground nests was significantly greater than for haystacks (Mann–Whitney $U$, $Z = -1.97$, $P = 0.049$; Table 2). Of the 29 nests monitored, 24 were haystack nests (6 in riverine wetlands, 9 in ponds and 9 in marshes).

**Table 2 Nest characteristics (mean ± SD) of Black-necked Cranes in Yanchiwan National Nature Reserve, Gansu, China.**

| Nest types | Length (cm) | Width (cm) | Height (cm) | Water depth (cm) | Nest success | Nest survival rate |
|---|---|---|---|---|---|---|
| Haystack ($n = 24$) | 114.92 ± 31.48 | 98.14 ± 16.88 | 19.29 ± 8.06 | 32.63 ± 15.05 | 66.67% | 78.26% |
| Ground ($n = 5$) | 107.60 ± 39.67 | 83.40 ± 7.60 | 12.00 ± 7.81 | 54.40 ± 24.52 | 60.00% | 80.00% |
| F/Z[a] or [b] | −1.22 | −2.26 | 3.42 | −1.97 | −0.28 | −0.04 |
| P | 0.224[ns] | 0.024[*] | 0.075[ns] | 0.049[*] | 0.779[ns] | 0.967[ns] |

Notes.
[ns]not significant.
[*]$p < 0.05$.
[a]One-way ANOVA test.
[b]Mann–Whitney $U$ test. Nests were considered as "successful" if at least one chick or more chicks were confirmed to have hatched. Nest survival rate was calculated as the percentage of the total number of successful nests from total number of nests initially found.

**Table 3 Nest parameters and site characteristics (mean ± SD) of three microhabitats and comparison between successful and unsuccessful nests of Black-necked Cranes in Yanchiwan National Nature Reserve, Gansu.**

| Variables | Pond ($n = 13$) | Marsh ($n = 10$) | Riverine wetland ($n = 6$) | P | Sn ($n = 23$) | Un ($n = 6$) | P |
|---|---|---|---|---|---|---|---|
| NL | 109.31 ± 25.32 | 115.60 ± 35.57 | 119.83 ± 44.09 | 0.882 | 111.43 ± 22.81 | 109.17 ± 17.44 | 0.957 |
| NW | 89.75 ± 8.92 | 103.05 ± 18.11 | 95.83 ± 23.54 | 0.122 | 100.48 ± 16.96 | 101.50 ± 11.96 | 0.891 |
| NH | 15.31 ± 6.24 | 18.80 ± 5.42 | 22.67 ± 13.89 | 0.195 | 18.26 ± 9.05 | 17.17 ± 5.46 | 0.781 |
| WD | 51.46 ± 15.66 | 21.00 ± 9.87 | 29.33 ± 5.24 | 0.000[***] | 41.35 ± 17.28 | 17.33 ± 7.23 | 0.003[**] |
| DL | 17.92 ± 6.24 | 4.40 ± 2.99 | 11.00 ± 5.51 | 0.000[***] | 13.35 ± 7.79 | 6.00 ± 5.40 | 0.039[*] |
| DH | 30.77% [a], 69.23[b] | 100.00% [a] | 100.00% [a] | 0.000[***] | 65.22 [a], 34.78% [b] | 83.33% [a], 16.67% [b] | 0.401 |
| DI | 15.38% [c], 84.62% [d] | 90.00% [c], 10.00% [d] | 83.33% [c], 16.67% [d] | 0.001[**] | 43.48% [c], 56.52% [d] | 100% [c] | 0.015[*] |
| WA | 100.00% [f] | 50.00% [e], 50.00% [f] | 16.67% [e], 83.33% [f] | 0.015[*] | 8.70% [e], 91.30% [f] | 66.67% [e], 33.33% [f] | 0.002[**] |
| NS | 92.31% | 60.00% | 83.33% | 0.170 | | | |

Notes.
NL, nest length (cm); NW, nest width (cm); NH, nest height (cm); WD, water depth (cm); DL, distance from nest to the nearest land (m); DH, distance from nest to the nearest hill (m); DI, disturbance; WA, water body area (m²); Sn, successful nests; Un, unsuccessful nests; NS, nest survival rate.
[a]Distance from nest to the nearest hill longer than 100 m.
[b]Distance from nest to the nearest hill shorter than 100 m.
[c]Strong disturbance.
[d]Weak disturbance.
[e]Water body area <500 m².
[f]Water body area >500 m².
[***]$p < 0.001$.
[**]$p < 0.01$.
[*]$p < 0.05$.

Five nests were ground nests (4 in ponds and 1 in marsh). Five of 6 unsuccessful nests were haystack nests.

## Nest site characteristics and their influences on nest survival

All nest distances to the nearest hills in riverine wetlands and marshes were longer than 100 m, but 9 pond nest site distances to the nearest hills were shorter than 100 m. Disturbance levels in most pond nest sites were inaccessible for predators or human but disturbance levels in most nests in marshes and in riverine wetlands were accessible (Table 3). All water body areas in pond nest sites and 83.3% of riverine wetland nest sites were greater than 500 m², but only half of nest sites in marshes were bigger than 500 m².

**Table 4 Principal component loadings through a rotation method for six characteristics of nest site selection by Black-necked Cranes in Yanchiwan National Nature Reserve, Gansu, China.**

| Variables | PC I | PC II |
|---|---|---|
| DI | −0.907 | −0.112 |
| WD | 0.827 | 0.248 |
| HT | −0.039 | −0.948 |
| DL | 0.736 | 0.441 |
| DH | −0.714 | −0.010 |
| WA | 0.550 | 0.549 |
| % of total variance | 56.08% | 16.07% |
| % of cumulative variance | 56.08% | 72.15% |

Notes.

DI, disturbance; WD, water depth; HT, habitat types; DL, distance from nest to the nearest land; DH, distance from nest to the nearest hill; WA, water body area.

Nests were differently distributed across three habitat types: ponds (43.83%), marshes (34.48%) and riverine wetlands (20.69%; Table 3). There were no significant differences among nest length, nest height and nest width in the three habitats (Table 3). However, the average nest water depth in ponds was significantly greater than for nests in riverine wetlands and marshes (Kruskal–Wallis, $\chi 2 = 16.02$, $P = 0.000$; Table 3). Nests in marshes were the widest, while the height and length of nests in the riverine wetlands were the biggest (Table 3).

All five nest site characteristics including water depth, distance to land, distance to the nearest hills and disturbance were significantly different among the three microhabitats (Table 3). Nest survival rate in ponds was the highest among the three microhabitats (Table 3) but there was no significant differences for nest survival rate among the three microhabitats (Kruskal–Wallis, $\chi 2 = 3.54$, $P = 0.170$, Table 3). Of habitat types for six unsuccessful nests, 4 were in ponds, 1 in marsh and 1 in a riverine wetland. Nest water depth of 6 unsuccessful nests was significantly less than for 23 successful nests (Mann–Whitney $U$, $Z = -2.94$, $P = 0.003$, Table 3). The level of disturbance for 5 unsuccessful nests was strong, significantly different than disturbance level for successful nests (Mann–Whitney $U$, $Z = -3.18$, $P = 0.015$, Table 3). Distances to the nearest hills of 5 unsuccessful nests were longer than 100 m, although there was no significant difference between unsuccessful and successful nest distances to hills (Mann–Whitney $U$, $Z = -0.84$, $P = 0.401$, Table 3). In addition, distances to the nearest land of unsuccessful nests were significantly shorter than the distances for successful nests (one-way ANOVA, $F_{1,27} = 4.68$, $P = 0.039$, Table 3). Water body area of successful nests was significantly larger than for unsuccessful nests (Mann–Whitney $U$, $Z = -3.07$, $P = 0.002$, Table 3).

## Nest site selection

Of the nest site characteristics, principal component 1 (PC1) appeared to account for 56.08% of the variance for six characteristics. Disturbance exhibited the highest influence on PC1 with 44.83% ($N = 13$ nests) for nest sites with weak disturbance and 82.76% of nest water depth larger than the average nest water depth of failed nests. PCII accounted for 16.07% and nest site habitats were highly correlated with PCII (Table 4). 44.83% of

nest sites were located in ponds, and 34.48% of nest sites were located in marshes while only 20.69% of nests were located in riverine wetlands (Table 3).

## DISCUSSION

### Population survey

Numbers of BNC and breeding pairs remained relatively stable at around 140 individuals and 40 pairs, respectively, during our continuous monitoring period. Our surveys indicated that the greatest number of BNC observed was on 15 July in 2015. However, the peak count of BNC at Longbao Wetland was 216 individuals on 25 April 2011 for that vital stopover site during migration (*Farrington & Zhang, 2013*). Eggs generally were laid from early May to mid-June in YCW while the first egg was laid on 30 April in Longbao Wetland (*Farrington & Zhang, 2013*). Recruitment is a vital element of avian population dynamics and is often considered to provide an estimation of population fluctuation over time (*Shaffer, 2004*). Average chick recruitment was 20.17%, higher than that for Zhigatse Prefecture (*Bishop, Tsamchu & Li, 2012*) and for Xinjiang (*Zhang et al., 2012*), but less than reported from Longbao Wetland (*Farrington & Zhang, 2013*). Fences in YCW keep livestock from private grassland, but BNC at Ruoergai were also influenced by presence of fences due to longer search time needed as the cranes flew from one patch to another (*Wu et al., 2009*). In our case, fences not only affected adult cranes, but also hurt chicks, which were unable to avoid the fences (Fig. S5). Nest survival rate has been identified as one of the most important components of recruitment (*Walker et al., 2005*). Twentythree of 29 nests in YCW were successful, which was similar to the nest survival rate in Longbao Wetland (*Farrington & Zhang, 2013*).

The nest density recorded was 0.9–1.2 pairs/km$^2$, which was clearly less than the values reported of 1.8 pairs/km$^2$, or 2.2 pairs/km$^2$ (*Dwyer et al., 1992*). This difference could perhaps be caused by lower wetland habitat suitability and food availability at the periphery of the range. Average crane neighboring nest distance in YCW coincided well with distance between neighboring nests reported from Qinghai province (*Lu, Yao & Liao, 1980*). Although territorial crane families appeared to occupy the same territories year after year, average space between nests in YCW in 2014 was much larger than for the other two years. This phenomenon might have resulted from drought in 2014, the monitoring year with less precipitation (33.5 mm; *Li et al., 2016*), leading to drying out of shallow ponds or marshes and limited nest site availability (Fig. S6). So the size of breeding territory area depended not only on breeding habitat status, but also on weather conditions and climate variability such as drought. Nest survival rate was the lowest in 2014 of the three monitoring years, similar to a report for Eurasian Crane (*Grus grus*) for which nesting success decreased when the distance between nests became too long (*Leito et al., 2005*).

### Reproductive performance

*Duangchantrasiri et al. (2016)* also believed cameras can provide quality data when quantity is not possible for small threatened populations. Our data, from infrared cameras used to monitor nest fate and the incubation process, provided unambiguous reproductive success data. In studies for some crane species, nest success, breeding success and hatching success

have all been determined (*Mukherjee, Borad & Parasharya, 2002*; *Ivey & Dugger, 2008*). Yet this study is the first for BNC to report and compare these three variables. No information seems to be available on the nest and hatching success in the BNC. So we only can compare our data with other closely related cranes. Average nest survival rate from 2013 to 2015 was 66%, lower than nest success (72%) for Greater Sandhill Crane (*Grus canadensis tabida*; *Ivey & Dugger, 2008*) and 71.43% for Indian Sarus Crane (Antigone antigone antigone; *Mukherjee, Borad & Parasharya, 2002*). As for hatching success, the figure 57% was lower than 62.5% reported for Indian Sarus Crane (*Mukherjee, Borad & Parasharya, 2002*). The average breeding success of 33% from 2013 to 2015 was slightly higher than breeding success of 25.74% for Indian Sarus Crane (*Mukherjee, Borad & Parasharya, 2002*).

## Foraging resource density and quantity

Habitat preferences for most species are associated with the density of tubers present, which differ remarkably among habitats (*Robinson & Sutherland, 1999*). BNC often foraged in mudflat and marsh but seldom in saline-alkali wet meadows at YCW, which can be explained by significantly different average tuber densities and fresh weight in the three foraging habitats (Fig. 5). Both tuber densities and fresh weight were the highest in mudflat of the three foraging habitats (Fig. 5), consistent with a study by *Luo et al. (2013)* who found that the weight of those growing in shallow water habitats was higher than for those growing on land or in deep water. BNC at YCW preferred foraging on mudflat while in northern Tibet, peat land with shallow water was their favorite foraging habitat (*Kuang et al., 2010*).

*Lu, Yao & Liao (1980)* found breeding pairs often present within 200 to 300 m of their nesting sites during the breeding period. *Farrington & Zhang (2013)* reported that breeding pairs did not shift their territories until just before leaving for autumn migration. Our data, however, from two satellite-tracked rescued chicks in YCW indicated that crane families shifted their roosting sites several times and settled 3.22 km (BNC-1) and 1.55 km (BNC-2) distant from their original nests. The distance and the territory shift between seasons depended positively on the chicks' flight ability in this study. *Bradter et al. (2007)* also reported one breeding White-naped Crane (*Antigone vipio*) pair permanently shifting to another wetland about 2 km from their original nest site, due to the wetland adjacent to the nest site being too small to provide enough foraging habitat for the family. This factor might explain shifts in core-use areas by BNC. Such shifts within core-use areas demonstrate that maintaining crane populations will demand the conservation of areas large enough to permit breeding BNC to change locations of habitat use, consistent with the proposal for White-naped Cranes that breeding pairs require at least 3 km distance away from their roosting or nest site (*Bradter et al., 2007*).

## Nest characteristics and their influence on nest survival

Cranes in YCW showed a strong preference for haystack nests (24/29), which were constructed directly in the water and required more mud and plant rhizomes and cost more energy (*Dwyer et al., 1992*). On the contrary, ground nests, which were constructed on pre-existing islands, were chosen in Tibet (*Dwyer et al., 1992*) and in Ruoergai Wetland (*Wu et al., 2009*). Nest material has been found to have some influence on nesting success,

such as hay with better mechanical or thermo regulatory characteristics for incubation (*Leito et al., 2005*). Our study, however, showed there was no significant difference in nest survival rate between haystack nests and ground nests (Table 2). We speculate that our limited sample may be the cause. Eighty percent of ground nests were located in ponds at YCW. Perhaps fewer ground nests were chosen due to the low availability of suitable islands in YCW. Nests in YCW were larger and built in deeper water than those reported elsewhere in China. BNC in YCW have to constantly maintain their nests at a certain height above water, resulting in bigger nests (*Wu et al., 2009*), because water level frequently fluctuates due to varying glacier meltwater supply.

## Nest site characteristics and their influences on nest survival

Disturbance, water depth and nest habitat type were likely to be limiting factors for nest site selection at YCW (Table 4). Successful nests had weak disturbance, significantly different from unsuccessful nests that had strong disturbance (Table 3). Untied dogs, livestock and humans are major disturbances in YCW, as also observed in Longbao Wetland (*Farrington & Zhang, 2013*). In contrast to no water depth differences among different microhabitats for nest selection in Ruoergai Wetland (*Wu et al., 2009*), water depth differences were found among different microhabitats in YCW, which might result from glaciers contributing to wetland water resources. *Jiao et al. (2014)* found water area and water depth were the main factors that influenced Hooded Crane (*Grus monacha*) nest site selection. They also pointed out that more water area can supply more food for chick cranes and water depth can protect cranes from some predators, similar to our results that larger water body area correlates with greater nest survival rate (*Jiao et al., 2014*).

Habitat preference may vary regionally. All 29 nests in YCW were located in wetlands. Almost half of the nests were situated in ponds. The cranes may select ponds due to their comparatively high nest survival rate (92.31%, Table 3), the deepest water depth, weakest disturbance level and biggest water body area of the three microhabitats. On the other hand, only 13 of 29 nest sites were placed in ponds, suggesting that selection of ponds for nesting was proportional to their availability in the landscape. *Bradter et al. (2005)* and *Wu et al. (2009)* also reported that availability limited occupation of advantageous nest sites. The greater use of marshes ($N = 10$ nests) in YCW, compared to Ruoergai, might be attributed to the greater prevalence of marshes (4.9% marsh vs 2.4% permanent pond + riverine wetland) in YCW and plenty of food compared to the saline-alkali wet meadows (Fig. 5). And foraging in marshes would reduce energy expenditure and searching time, which are limiting resources for incubating parents (*Bradter et al., 2007*), leading to a high nest survival rate especially in alpine altitudes (*Macdonald et al., 2015*). Furthermore, the sticky mud hinders predators or humans in approaching nests in marshes (*Dwyer et al., 1992*). To generalize, it could be stated that cranes are well adapted to breed in different types of wetlands available in an area, but prefer pond and marsh habitats and avoid, when possible, saline habitats.

The Yanchiwan National Nature Reserve is one of the most important breeding sites for the vulnerable Black-necked Crane. We conclude that the Black-necked Crane breeds in several types of wetland in YCW, with the favorite nesting habitat being marshes especially

ponds. They avoid saline wetlands for foraging. Water depth, water body area, distance to land and disturbance level were related to nest survival but cranes based nest site selection on habitat type, disturbance and water depth. Identifying factors that determine the BNC nest site selection at the northern range limit is not only a simple ecological question but also an important conservation issue. To our knowledge, this study is the first to document breeding Black-necked Cranes shifting their territory, which we documented by satellite tracking. Ecologists and conservation biologists should view habitat patches in terms of food distribution and abundance, not just nest and nest site requirements.

## ACKNOWLEDGEMENTS

We are grateful to all people who helped with the fieldwork in Danghe Wetland, to Lanzhou University for permission to carry out field studies, to Dr. Bo Du for statistical assistance, Jim Harris and Mary Bishop for their invaluable suggestions and editing on the original manuscript and two anonymous referees for their constructive comments.

### Funding

This research was financially supported by a grant from the State Key Laboratory of Genetic Resources and Evolution, Kunming Institute of Zoology, Chinese Academy of Sciences (GREKF13-12). This work was further supported by Yanchiwan National Nature Reserve. The funders had no role in study design, data collection and analysis, decision to publish, or preparation of the manuscript.

### Grant Disclosures

The following grant information was disclosed by the authors:
State Key Laboratory of Genetic Resources and Evolution.
Kunming Institute of Zoology.
Chinese Academy of Sciences: GREKF13-12.
Yanchiwan National Nature Reserve.

### Competing Interests

The authors declare there are no competing interests.

### Author Contributions

- Lixun Zhang conceived and designed the experiments, performed the experiments, analyzed the data, contributed reagents/materials/analysis tools, wrote the paper, reviewed drafts of the paper.
- Bei An conceived and designed the experiments, wrote the paper, reviewed drafts of the paper.
- Meilin Shu analyzed the data, prepared figures and/or tables, reviewed drafts of the paper.
- Xiaojun Yang conceived and designed the experiments, contributed reagents/materials/analysis tools, reviewed drafts of the paper.

## Animal Ethics

The following information was supplied relating to ethical approvals (i.e., approving body and any reference numbers):

Lanzhou University Institutional Animal Care and Use Committee.

Approval numbers: SCXK-GAN-2013-0003.

## Field Study Permissions

The following information was supplied relating to field study approvals (i.e., approving body and any reference numbers):

Authority of the Forestry Department of Gansu Province.

Approval number: 201009.

## Data Availability

The raw data has been supplied as a Supplementary File.

## Supplemental Information

Supplemental information for this article can be found online at http://dx.doi.org/10.7717/peerj.2939#supplemental-information.

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
