# Peer review of "Nest-site selection, reproductive ecology and shifts within core-use areas of Black-necked Cranes at the northern limit of the Tibetan Plateau"

_PeerJ, doi:10.7717/peerj.2939_

## Round 0.1 · original submission · Major Revisions

As you can see, the reviewers have made many useful comments. Please revise your manuscript accordingly.

Reviewer 1 ·

Basic reporting

The article meets the PeerJ standards except the written English could be improved - see 'General comments for the authors' for some details.

Experimental design

Experimental Design and Validity of the Findings correspond to each other and accordingly comments on the two parts are hardly dividable. I will put my comments on the two parts here together.

In this manuscript the authors investigated population dynamics, breeding habitat selection and reproductive success of Black-necked Crane at the northern limit of the Tibetan Plateau. This manuscript is important as the species is threatened and the study area is important, and this is the first study investigating this topic. In this study the authors collected impressive and fruitful amount of data of the basic ecology of BNC, which shows the potential value of this paper. However, I think the authors could greatly improve the organization of the data, which currently blurred the focus of the paper. I highly expect that the manuscript would greatly improve by rethinking of the hypotheses and data analyses. In the current submission the authors mainly used some statistics to compare some parameters (e.g. nest parameters, success rate). They are correct to distinguish the parametric/non-parametric statistics according to the data distribution. However, the comparisons are not optimal, or not correct without considering other factors. Thus linear models might be useful in this case, to incorporate various variables to understand how nest parameters, nest selection could affect nest success/survival. If possible, food quality/quantity should be also included in the model to better link them with nest survival/recruitment. The result of the linear models could also help to explain the results of nest site selection.
Moreover, the authors should revise the Introduction & Methods & Results carefully to avoid any mix/inconsistency/mismatch between them, please see specific comments below. The satellite tracking of this species is important and informative, but in this manuscript this part of results did not integrate well into the story.

Validity of the findings

See the box above.

Additional comments

L80: has -> have
L85: Please cite IUCN/BirdLife with its Current IUCN Red List category.
L86: “due to widespread wetland loss and agricultural development” – need reference.
L91: “general ecology” – please clarify
L92: “nest parameters” – please clarify.
L100: Please define ‘LCW’ for the first use.
L100-104: Unclear, please rephrase.
L112: Please rephrase
L113: Name of the reserve is missing.
L127: Please include a map to show the location of nature reserve, survey routes and nest locations if possible. Survey date and routes covered in each survey should also be listed.
L136-141: Here the satellite tracking and movement analyses are not mentioned in Introduction, abstract or title, while they should be, although the sample size is small, it provided novel and useful results. Also, if the authors decide to include these home range analyses, methods of these movement/home range analyses should be described in detail; the transmitter and harnessing should be explained in more detail.
L146: How was disturbance measured?
L168: are -> were
L168: As it was stated “the diets of BNC consist primarily of roots and tubers, insects, snails, shrimps, fish, small birds and rodents”, can sampled tuber in this study represent food resource/quality of BNC in this area?
L175: The parameter ‘nest density’ should be introduced in Introduction. And also please justify the method of measuring nest density with reference, my concern is that a ‘minimum’ value seems has little representativeness of a population. And this is also inconsistent with L231-L236, where nest per km^2 and average nest distance were used.
L177: yes -> so
L179: Delete “Three elements, which were …”.
L199: In corresponding section in Methods, please describe how age was determined.
L200-201: please describe in Methods how territories and nest initiation were determined; ‘with a median date of April 18’ looks odd.
L204: Please provide a full list of survey data for the three study years.
L223: “tended to build their nests later than pairs building haystack nests” - do you have any supporting data?
L234-235: Although technically correct, I would remove the statement ‘… insignificantly higher’ since either the statistics or the percentage value didn’t provide any support.
L235: May use some statistics, such as Chi-square tests, to compare nest success between haystack and ground.
L239-L243: Should move to Methods section.
L214&238: Since both nest characteristics and nest site could affect nest survival/success, I would combine the analyses in the two sections with (generalized) linear models to investigate these factors on nest survival/success
L269: Please describe the methods used in this section in Methods.
L283-287: Results in this section is inconsistent to the objective (4) claimed in Introduction L107-108.
L381: Unify the format of this subheading.
L381-409: This part is little bit strange as the main text did not investigate much foraging/roosting habitat selection.

Table 1: needs more details about the statistics method used here, the definition of characteristics here.

Reviewer 2 ·

Basic reporting

A lot of revision needs to be done in presentation of language. Terms like breeding couples and chick born needs to be replaced with proper scientific words. As for example instead of chicks born in line 297 it should be chicks hatched. In line 139 remove the words like luckily.

The introduction and background section needs a through review.

Please recheck the line 123 which mentions snowfall in any month.

In line 283 very preliminary information is given on foraging habitat. Please give much more quantified information on foraging.

Experimental design

The structure of the paper confirms the PeerJ standard.

The figures in the paper are quite relevant but need proper labelling and description.

Validity of the findings

Data collected is of high quality but needs much more comparisons and interpretation in the concluding section.

This is a original research work with many conservation implications for the species. The conclusion section needs to be throughly reviewed as currently the link to the original research questions is not properly interpreted. This needs to be linked to the research questions with much more clarity.

Additional comments

I generally believe that this is a good work and a lot of effort has been put in the field.

I recommend the manuscript for publication after review as per the points mentioned above and hope that my specific points might improve the manuscript.

---

## Round 0.2 · Minor Revisions

The reviewers have made only minor suggestions, but please address them in your revision. (That revision should not take 40 days, as per the PeerJ boilerplate — they are all simple.) I expect to accept your paper when you have made these revisions and to do so without sending the paper out for further review.

I urge you to check the writing, using a programme such as Grammarly, or better (or both!) to seek the help of a native English speaker. PeerJ does not copy edit manuscripts so what you send in what will be published.

Finally, thank you for submitting this to PeerJ. I'm delighted that we are attracting so many good papers on conservation and, especially so, to see work on this exiting species, which I've been lucky to see in the wild in China.

Reviewer 1 ·

Basic reporting

Although the English have been greatly improved, this manuscript still need be polished by a native English speaker to improve linguistics.

Experimental design

No Comments

Validity of the findings

No Comments

Additional comments

It's glad to see this manuscript "Nest-site selection, reproductive ecology and shifts in coreuse areas of Black-necked Cranes at the northern limit of the Tibetan Plateau" by Zhang et al. again. Many effors have been invested to improve the quality of this MS such as in method part and data process. The author also tried to deal with the data using generelize mixed linear model although failed. However, the language still need to be polised by a native english speaker as there are still some errors. Here i attached my tiny comments below:
Line 37 delete “preferences for”
Line 38 But cranes…. May be rephrased to “However, nest site selection of BNC was determined by habitat type……”
Line 42 change to “Shifts within core-use areas derived from……..”
Line 43 “are” instead of “be”
Line 93 add “however” before “only”
Line 125 Rephrase: In all, 23 times surveys were conducted…… and this first sentence should be removed to the results part
Line 148 by Higuchi et al. (2004)
Line 172 the tubers of which species?
Line 176 density and weight are not “quality” but probably “quantity”
Line 186 add “as” before “a”
Line 198 what is nonparametric t-test? KW and MW U test are not t-test but nonparametric test!
Line 203 what are the three nesting habitat, should be explained in method part or somewhere else.
Line 204 assessed changed to compared
Line 221 three monitoring years
Line 228-229 not clear, sentence need rephrase
Line 233 to be consistent, I think a space should be added before and after a mathematical notation. Check the whole MS.
Line 250 would change to tend to
Line 267-268 sentence need be modified.
Line 295 Forty-four point eight three? Use 44.83%
Line 302 monitoring
Line 321-322 sentence need rephrase
Line 336 “)” loss
Line 343 between changes to among
Line 347 for sedges changes to those
Line 372-372 is BNC larger or the nest?
Line 401 “.” loss
Use figure to illustrate the results present in Table 2 and label the differences among habitat.

Reviewer 2 ·

Basic reporting

There is a lot of improvement in the manuscript and most of the comments have been incorporated.
However, still in line 99 use the word northern most instead of most northern. In line 99 use snowfall can be recorded any time instead of snow can fall.in line 295 write as 44.83% instead of words. In line 332 use word hatching success instead of hatch success.

Experimental design

This study design is fine and good quality data collected through through investigation has been analysed .

The methods described are fine.

Validity of the findings

I congratulate the authors for all their hard work on data collection in such a tough terrain. However one factual error in lines 159 and 160 needs to be corrected. The finding that the researchers visited nests when both parents leave nest for feeding is incorrect. Cranes specially black-necked crane never leaves the nest unattended. If one parent goes for feeding, the other takes care of the nest. Please check this and modify your result accordingly.

Additional comments

Overall this is a very good paper and has a lot of interesting and useful information which has lot of conservation implications.

---

## Round 0.3 · accepted · Accept

Thank you for sending your revision so quickly!